# SupportNet: solving catastrophic forgetting in class incremental learning with support data

## Abstract

A plain well-trained deep learning model often does not have the ability to learn new knowledge without forgetting the previously learned knowledge, which is known as *catastrophic forgetting*. Here we propose a novel method, Support-Net, to efficiently and effectively solve the catastrophic forgetting problem in the class incremental learning scenario. SupportNet combines the strength of deep learning and support vector machine (SVM), where SVM is used to identify the support data from the old data, which are fed to the deep learning model together with the new data for further training so that the model can review the essential information of the old data when learning the new information. Two powerful consolidation regularizers are applied to stabilize the learned representation and ensure the robustness of the learned model. We validate our method with comprehensive experiments on various tasks, which show that SupportNet drastically outperforms the state-of-the-art incremental learning methods and even reaches similar performance as the deep learning model trained from scratch on both old and new data.

## 1 Introduction

Since the breakthrough in 2012 (Krizhevsky et al., 2012), deep learning has achieved great success in various fields (LeCun et al., 2015; Silver et al., 2016; Sutskever et al., 2014; He et al., 2016; Alipanahi et al., 2015; Li et al., 2018b; Dai et al., 2017). However, despite its impressive achievements, there are still several bottlenecks related to the practical part of deep learning waiting to be solved (Papernot et al., 2016; Lipton, 2016; Kemker et al., 2017). One of those bottlenecks is *catastrophic forgetting* (Kemker et al., 2017), which means that a well-trained deep learning model tends to completely forget all the previously learned information when learning new information (McCloskey & Cohen, 1989). That is, once a deep learning model is trained to perform a specific task, it cannot be trained easily to perform a new similar task without affecting the original task's performance dramatically. Unlike human and animals, deep learning models do not have the ability to continuously learn over time and different datasets by incorporating the new information while retaining the previously learned experience, which is known as *incremental learning*.

Two major theories have been proposed to explain human's ability to perform incremental learning. The first theory is Hebbian learning (Hebb, 1949) with homeostatic plasticity (Zenke et al., 2017), which suggests that human brain's plasticity will decrease as people learn more knowledge to protect the previously learned information. The second theory is the complementary learning system (CLS) theory (Mcclelland et al., 1995; OReilly et al., 2014), which suggests that human beings extract high-level structural information and store the high level information in a different brain area while retaining episodic memories. Inspired by the above two major neurophysiological theories, people have proposed a number of methods to deal with catastrophic forgetting. The most straightforward and pragmatic method to avoid catastrophic forgetting is to retrain a deep learning model completely from scratch with all the old data and new data (Parisi et al., 2018). However, this method is proved to be very inefficient (Parisi et al., 2018). Moreover, the new model learned from scratch may share very low similarity with the old one, which results in poor learning robustness. In addition to the straightforward method, there are three categories of methods. The first category is the regularization approach (Kirkpatrick et al., 2017; Li & Hoiem, 2016; Jung et al., 2016), which is inspired by

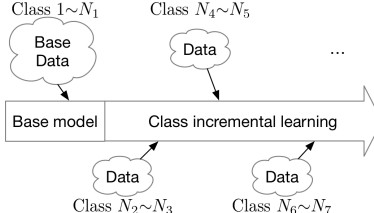

Figure 1: Illustration of class incremental learning. After we train a base model using all the available data at a certain time point (e.g., classes $1 \sim N_1$), new data belonging to new classes may continuously appear (e.g., classes $N_2 \sim N_3$, classes $N_4 \sim N_5$, etc) and we need to equip the model with the ability to handle the new classes.

the plasticity theory (Benna & Fusi, 2016). The core idea of such methods is to incorporate the plasticity information of the neural network model into the loss function to prevent the parameters from varying significantly when learning new information. These approaches are proved to be able to protect the consolidated knowledge (Kemker et al., 2017). However, due to the fixed size of the neural network, there is a trade-off between the performance of the old and new tasks (Kemker et al., 2017). The second class uses dynamic neural network architectures (Rebuffi et al., 2016; Rusu et al., 2016; Lopez-Paz & Ranzato, 2017). To accommodate the new knowledge, these methods dynamically allocate neural resources or retrain the model with an increasing number of neurons or layers. Intuitively, these approaches can prevent catastrophic forgetting but may also lead to scalability and generalization issues due to the increasing complexity of the network (Parisi et al., 2018). The last category utilizes the dual-memory learning system, which is inspired by the CLS theory (Hinton & Plaut, 1987; Lopez-Paz & Ranzato, 2017; Gepperth & Karaoguz, 2016). Most of these systems either use dual weights or take advantage of pseudo-rehearsal, which draw training samples from a generative model and replay them to the model when training with new data. However, how to build an effective generative model remains a difficult problem.

Recent researches on the optimization and generalization of deep neural networks suggested the potential relationship between deep learning and SVM (Soudry et al., 2017; Li et al., 2018a). Based on that idea, we propose a novel and easy-to-implement method to perform class incremental deep learning efficiently when encountering data from new classes (Fig. 1). Our method maintains a support dataset for each old class, which is much smaller than the original dataset of that class, and shows the support datasets to the deep learning model every time there is a new class coming in so that the model can "review" the representatives of the old classes while learning new information. Although this *rehearsal* idea is not new (Rebuffi et al., 2016), our method is innovative in the sense that we show how to select the support data in a systematic and generic way to preserve as much information as possible. We demonstrate that it is more efficient to select the support vectors of an SVM, which is used to approximate the neural network's last layer, as the support data, both theoretically and empirically. Meanwhile, since we divide the network into two parts, the last layer and all the previous layers, in order to stabilize the learned representation of old data before the last layer and retain the performance for the old classes, following the idea of the Hebbian learning theory, we utilize two *consolidation regularizers*, to reduce the plasticity of the deep learning model and constrain the deep learning model to produce similar representation for old data. The framework of our method is show in Fig. 2. In summary, this paper has the following main contributions:

- We propose a novel way of selecting support data through the combination of deep learning and SVM, and demonstrate its efficiency with comprehensive experiments on various tasks.

- We propose a novel regularizer, namely, *consolidation regularizer*, which stabilizes the deep learning network and maintains the high level feature representation of the old information.

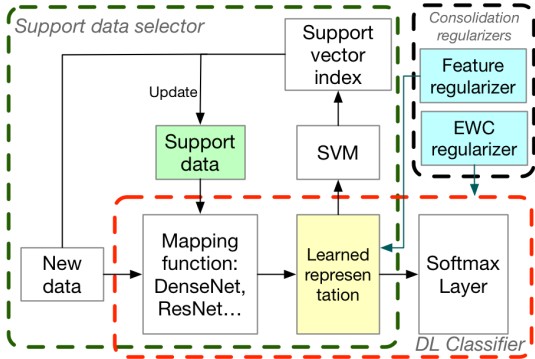

Figure 2: Overview of our framework. The basic idea is to incrementally train a deep learning model efficiently using the new class data and the support data of the old classes. We divide the deep learning model into two parts, the mapping function (all the layers before the last layer) and the softmax layer (the last layer). Using the learned representation produced by the mapping function, we train an SVM, with which we can find the support vector index and thus the support data of old classes. To stabilize the learned representation of old data, we apply two novel consolidation regularizers to the network.

## 2 METHODS

### 2.1 DEEP LEARNING AND SVM

In this subsection, we will show what data is more important for deep neural network model training. Following the setting in Soudry et al. (2017); Li et al. (2018a), let us consider a dataset $\{\mathbf{x}_n, \tilde{\mathbf{y}}_n\}_{n=1}^N$, with $\mathbf{x}_n \in \mathbb{R}^D$ being the feature, and $\tilde{\mathbf{y}}_n \in \mathbb{R}^K$ being the one-hot encoding of the label. $K$ is the total number of classes and $N$ is the size of the dataset. Denote the input of the last layer (the learned representation) as $\boldsymbol{\delta}_n \in \mathbb{R}^T$ for $\mathbf{x}_n$. We use $\mathbf{W}$ to denote the parameter of the last layer and define $\mathbf{z}_n = \mathbf{W}\boldsymbol{\delta}_n$. After applying softmax activation function to $\mathbf{z}_n$, we obtain the output of the whole deep neural network for the input $\mathbf{x}_n$ as $\mathbf{o}_n$. Consequently, we have:

$$o_{n,i} = \frac{\exp(z_{n,i})}{\sum_{k=1}^K \exp(z_{n,k})} = \frac{\exp(\mathbf{W}_{i,:}\boldsymbol{\delta}_n)}{\sum_{k=1}^K \exp(\mathbf{W}_{k,:}\boldsymbol{\delta}_n)}. \tag{1}$$

For deep learning, we usually use the cross-entropy loss as the loss function:

$$L = -\frac{1}{N}\sum_{n=1}^N \sum_{k=1}^K \tilde{y}_{n,k} \log(o_{n,k}), \tag{2}$$

Consider the negative gradient of the loss function on $w_{j,i}$ (the derivation of Equation (3) can be referred to Section A in the Appendices):

$$\begin{aligned} -\frac{\partial L}{\partial w_{j,i}} &= \frac{1}{N}\sum_{n=1}^N (\tilde{y}_{n,i} - o_{n,i})\delta_{n,j} \\ &= \frac{1}{N}\sum_{n=1}^N (\tilde{y}_{n,i} - \frac{\exp(\mathbf{W}_{i,:}\boldsymbol{\delta}_n)}{\sum_{k=1}^K \exp(\mathbf{W}_{k,:}\boldsymbol{\delta}_n)})\delta_{n,j}, \end{aligned} \tag{3}$$

according to Soudry et al. (2017); Li et al. (2018a), after the learned representation becoming stable, the last weight layer will converge to the SVM solution. That is, we can write $\mathbf{W} = a(t)\hat{\mathbf{W}} + \mathbf{B}(t)$, where $\hat{\mathbf{W}}$ is the corresponding SVM solution; $t$ represent the $t$-th iteration of SGD; $a(t) \to \infty$ and $\mathbf{B}(t)$ is bounded. Thus, Equation (3) becomes:

$$-\frac{\partial L}{\partial w_{j,i}} = \frac{1}{N}\sum_{n=1}^N (\tilde{y}_{n,i} - \frac{\exp(a(t)\hat{\mathbf{W}}_{i,:}\boldsymbol{\delta}_n)\exp(\mathbf{B}(t)_{i,:}\boldsymbol{\delta}_n)}{\sum_{k=1}^K \exp(a(t)\hat{\mathbf{W}}_{k,:}\boldsymbol{\delta}_n)\exp(\mathbf{B}(t)_{k,:}\boldsymbol{\delta}_n)})\delta_{n,j}. \tag{4}$$

Since the candidate value of $\tilde{y}_{n,i}$ is $\{0,1\}$ and if $\tilde{y}_{n,i} = 0$, that term in Equation (2) does not contribute to the loss. Only when $\tilde{y}_{n,i} = 1$ can the data contribute the loss and thus the gradient. Under that circumstance, since $a(t) \to \infty$, only the data with the smallest exponential nominator can contribute to the gradient. Those data are precisely the ones with the smallest margin $\hat{\mathbf{W}}_{i,:}\boldsymbol{\delta}_n$, which are the support vectors, for class $i$.

## 2.2 SUPPORT DATA SELECTOR

According to Sirois et al. (2008); Pallier et al. (2003), even human beings, who are proficient in incremental learning, cannot deal with catastrophic forgetting perfectly. On the other hand, a common strategy for human beings to overcome forgetting during learning is to review the old knowledge frequently (Murre & Dros, 2015). Actually, during reviewing, we usually do not review all the details, but rather the important ones, which are often enough for us to grasp the knowledge. Inspired by this, we design the support dataset and the review training process. During incremental learning, we maintain a support dataset for each class, which is fed to the model together with the new data of the new classes. In other words, we want the model to review the representatives of the previous classes when learning new information.

The main question is thus how to build an effective support data selector to construct such support data, which we denote as $\{\mathbf{x}_n^S, \tilde{\mathbf{y}}_n^S\}_{n=1}^{N_S}$. According to the discussion in Section 2.1, we know that the data corresponding to the support vectors in SVM solution contribute more to the deep learning model training. Based on that, we obtain the high level feature representations of the original input using deep learning mapping function and train an SVM classifier with these features. By performing the SVM training, we detect the support vectors from each class, which are of crucial importance for the deep learning model training. We define the original data which correspond to these support vectors as the *support data candidates*, which we denote as $\{\mathbf{x}_n^{SV}, \tilde{\mathbf{y}}_n^{SV}\}_{n=1}^{N_{SV}}$. If the required number of preserved data is smaller than that of the support vectors, we will sample support data candidates to obtain the required number. Formally:

$$\{\mathbf{x}_n^S, \tilde{\mathbf{y}}_n^S\}_{n=1}^{N_S} \subset \{\mathbf{x}_n^{SV}, \tilde{\mathbf{y}}_n^{SV}\}_{n=1}^{N_{SV}}. \tag{5}$$

Denote the new coming data as $\{\mathbf{x}_n^{new}, \tilde{\mathbf{y}}_n^{new}\}_{n=1}^{N_{new}}$, we have the new training data for the model as:

$$\{\mathbf{x}_n^S, \tilde{\mathbf{y}}_n^S\}_{n=1}^{N_S} \cup \{\mathbf{x}_n^{new}, \tilde{\mathbf{y}}_n^{new}\}_{n=1}^{N_{new}}, \tag{6}$$

## 2.3 CONSOLIDATION REGULARIZERS

Since the support data selection depends on the high level representation produced by the deep learning layers, which are fine tuned on new data, the old data feature representations may change over time. As a result, the previous support vectors for the old data may no longer be support vectors for the new data, which makes the support data invalid (here we assume the support vectors will remain the same as long as the representations are largely fixed, which will be discussed in more details in Section 4.2). To solve the issue, we add two consolidation regularizers to consolidate the learned knowledge: the feature regularizer, which forces the model to produce fixed representation for the old data over time, and the EWC regularizer, which consolidates the important weights that contribute to the old class classification significantly into the loss function.

### 2.3.1 FEATURE REGULARIZER

We add the following feature regularizer into the loss function to force the mapping function to produce fixed representation for old data. Following the setting in Section 2.1, $\boldsymbol{\delta}_n$ depends on $\phi$, which is the parameters of the deep learning mapping function. The feature regularizer is defined as:

$$R_f(\phi) = \sum_{n=1}^{N_S} \|\boldsymbol{\delta}_n(\phi_{new}) - \boldsymbol{\delta}_n(\phi_{old})\|_2^2, \tag{7}$$

where $\phi_{new}$ is the parameters for the deep learning architecture trained with the support data from the old classes and the new data from the new class(es); $\phi_{old}$ is the parameters for the mapping function of the old data; and $N_s$ is the number of support data.

This regularizer requires the model to preserve the feature representation produced by the deep learning architecture for each support data, which could lead to potential memory overhead. However, since it operates on a very high level representation, which is of much less dimensionality than the original input, the overhead is neglectable.

### 2.3.2 EWC REGULARIZER

According to the Hebbian learning theory, after learning, the related synaptic strength and connectivity are enhanced while the degree of plasticity decreases to protect the learned knowledge. Guided by this neurophysiological theory, the EWC regularizer (Kirkpatrick et al., 2017) was designed to consolidate the old information while learning new knowledge. The core idea of this regularizer is to constrain those parameters which contribute significantly to the classification of the old data. Specifically, the more a certain parameter contributes to the previous classification, the harder constrain we apply to it to make it unlikely to be changed. That is, we make those parameters that are closely related to the previous classification less "plastic". In order to achieve this goal, we calculate the Fisher information for each parameter, which measures its contribution to the final prediction, and apply the regularizer accordingly.

Formally, the Fisher information for the parameters $\theta = \{\phi, \mathbf{W}\}$ can be calculated as:

$$
\begin{aligned}
F(\theta) &= E[(\frac{\partial}{\partial\theta}\log f(X;\theta))^2|\theta] \\
&= \int(\frac{\partial}{\partial\theta}\log f(x;\theta))^2 f(x;\theta)dx,
\end{aligned}
\tag{8}
$$

where $f(x;\theta)$ is the functional mapping of the entire neural network.

The EWC regularizer is defined as follows:

$$
R_{ewc}(\theta) = \sum_i F(\theta_{old_i})(\theta_{new_i} - \theta_{old_i})^2,
\tag{9}
$$

where $i$ iterates over all the parameters of the model.

There are two major benefits of using the EWC regularizer in our framework. Firstly, the EWC regularizer reduces the "plasticity" of the parameters that are important to the old classes and thus guarantees stable performance over the old classes. Secondly, by reducing the capacity of the deep learning model, the EWC regularizer prevents overfitting to a certain degree. The function of the EWC regularizer could be considered as changing the learning trajectory pointing to the region where the loss is low for both the old and new data.

### 2.3.3 LOSS FUNCTION

After adding the feature regularizer and the EWC regularier, the loss function becomes:

$$
\tilde{L}(\theta) = L + \lambda_f R_f(\phi) + \lambda_{ewc} R_{ewc}(\theta),
\tag{10}
$$

where $\lambda_f$ and $\lambda_{ewc}$ are the coefficients for the feature regularizer and the EWC regularizer, respectively.

After plugging Eq. (2), (7) and (9) into Eq. (10), we obtain the regularized loss function:

$$
\begin{aligned}
\tilde{L}(\theta) = &-\frac{1}{N_S + N_{new}}\sum_{n=1}^{N_S+N_{new}}\sum_{k=1}^{K_t}\tilde{y}_{n,k}\log(o_{n,k})+ \\
&\sum_{n=1}^{N_S}\|\boldsymbol{\delta}_n(\phi_{new}) - \boldsymbol{\delta}_n(\phi_{old})\|_2^2 + \\
&\sum_i \lambda_{ewc}(\theta_{new_i} - \theta_{old_i})^2 \int(\frac{\partial}{\partial\theta_{new}}\log f(x;\theta_{new}))^2 f(x;\theta_{new})dx,
\end{aligned}
\tag{11}
$$

where $K_t$ is the total number of classes at the incremental learning time point $t$.

## 2.4 SUPPORTNET

Combining the deep learning model, which consists of the deep learning architecture mapping function and the final fully connected classification layer, the novel support data selector, and the two consolidation regularizers together, we propose a highly effective framework, SupportNet (Fig. 2), which can perform class incremental learning without catastrophic forgetting. Our framework can resolve the catastrophic forgetting issue in two ways. Firstly, the support data can help the model to review the old information during future training. Despite the small size of the support data, they can preserve the distribution of the old data quite well, which will be shown in Section 4.1. Secondly, the two consolidation regularizers consolidate the high level representation of the old data and reduce the plasticity of those weights, which are of vital importance for the old classes.

## 3 RESULTS

### 3.1 DATASETS

During our experiments, we used six datasets: (1) MNIST, (2) CIFAR-10 and CIFAR-100, (3) Enzyme function data (Li et al., 2018c), (4) HeLa (Boland & Murphy, 2001) and (5) BreakHis (Spanhol et al., 2016). MNIST, CIFAR-10 and CIFAR-100 are commonly used benchmark datasets in the computer vision field. MNIST consists of 70K 28*28 single channel images belonging to 10 classes. CIFAR-10 contains 60K 32*32 RGB images belonging to 10 classes, while CIFAR-100 is composed of the same images but the images are further classified into 100 classes. The latter three datasets are from bioinformatics. Enzyme function data[1] is composed of 22,168 low-homologous enzyme sequences belonging to 6 classes. The HeLa dataset[2] contains around 700 512*384 grayscale images for subcellular structures in HeLa cells belonging to 10 classes. BreakHis[3] is composed of 9,109 microscopic images of the breast tumor tissue belonging to 8 classes. Each image is a 3-channel RGB image, whose dimensionality is 700 by 460.

### 3.2 COMPARED METHODS

We compared our method with numerous methods. We refer the first method as the "All Data" method. When data from a new class appear, this method trains a deep learning model from scratch for multi-class classification, using all the new and old data. It can be expected that this method should have the highest classification performance. The second method is the iCaRL method (Rebuffi et al., 2016), which is the state-of-the-art method for class incremental learning in computer vision field Kemker et al. (2017). The third method is EWC . The fourth method is the "Fine Tune" method, in which we only use the new data to tune the model, without using any old data or regularizers. The fifth method is the baseline "Random Guess" method, which assigns the label of each test data sample randomly without using any model. We also compared with a number of recently proposed methods, including three versions of Variational Continual Learning (VCL) methods (Nguyen et al., 2018), Deep Generative Replay (DGR) (Shin et al., 2017), Gradient Episodic Memory (GEM) (Lopez-Paz et al., 2017), and Incremental Moment Matching (IMM) (Lee et al., 2017) on MNIST. In terms of the deep learning architecture, for the enzyme function data, we used the same architecture from Li et al. (2018c). As for the other datasets, we used the residual network with 32 layers. Regarding the SVM in SupportNet framework, based on the result from Soudry et al. (2017); Li et al. (2018a), we used linear kernel.

### 3.3 PERFORMANCE COMPARISON

For all the tasks, we started with binary classification. Then each time we incrementally gave data from one or two new classes to each method, until all the classes were fed to the model. For enzyme data, we fed one class each time. For the other five datasets, we fed two classes in each round. Fig. 3 shows the accuracy comparison on the multi-class classification performance of different methods, over the six datasets, along the incremental learning process.

---

[1]http://www.cbrc.kaust.edu.sa/DEEPre/dataset.html

[2]http://murphylab.web.cmu.edu/data/2DHeLa

[3]https://web.inf.ufpr.br/vri/breast-cancer-database/

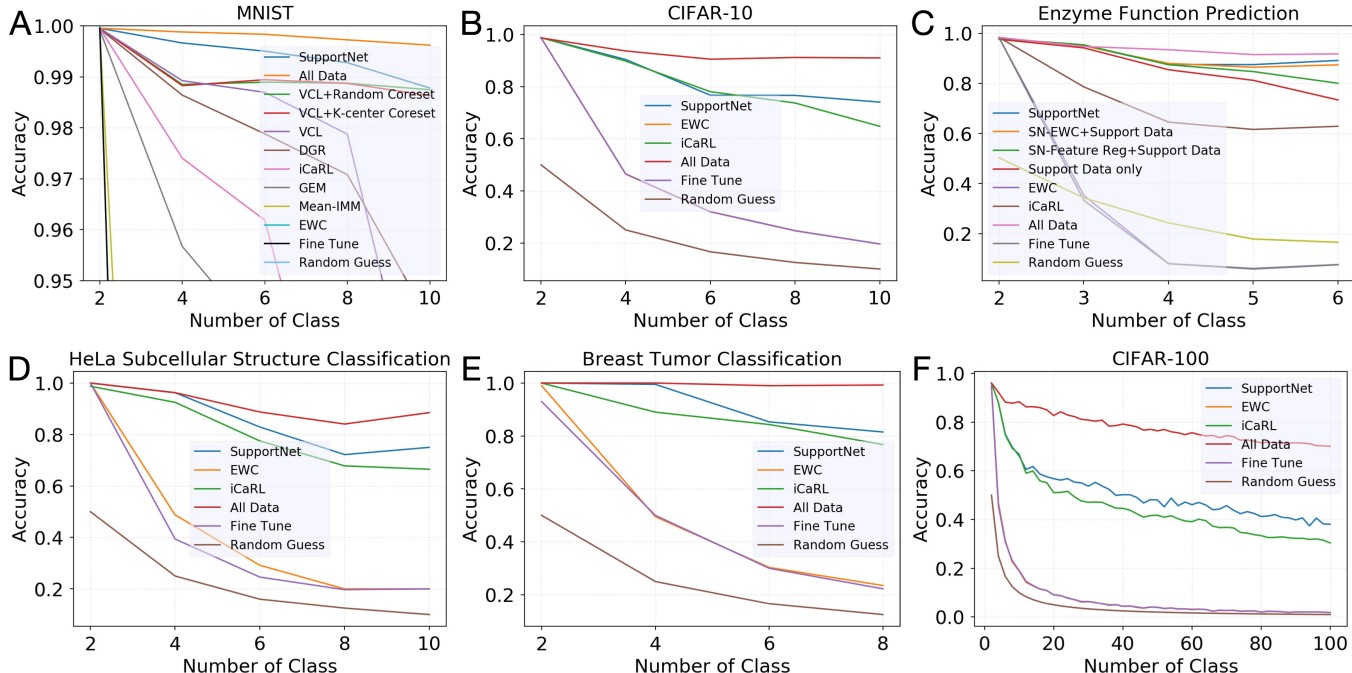

Figure 3: Main results. (A)-(F): Performance comparison between SupportNet and the competing methods on the six datasets in terms of accuracy. For the SupportNet and iCaRL methods, we set the support data (examplar) size as 2000 for MNIST, CIFAR-10, CIFAR-100 and enzyme data, 80 for the HeLa dataset, and 1600 for the breast tumor dataset.

As expected, the "All Data" method has the best classification performance because it has access to all the data and retrains a brand new model each time. The performance of this "All Data" method can be considered as the empirical upper bound of the performance of the incremental learning methods. All the incremental learning methods have performance decrease to different degrees. EWC and "Fine Tune" have quite similar performance which drops quickly when the number of classes increases. The iCaRL method is much more robust than these two methods. In contrast, SupportNet has significantly better performance than all the other incremental learning methods across the five datasets. In fact, its performance is quite close to the "All Data" method and stays stable when the number of classes increases for the MNIST and enzyme datasets. On the MNIST dataset, VCL with K-center Coreset can also achieve very impressive performance. Nevertheless, SupportNet can outperform it along the process. Specifically, the performance of SupportNet has less than 1% on MNIST and 5% on enzyme data difference compared to that of the "All Data" method. We also show the importance of SupportNet's components in Fig. 3 (C). As shown in the figure, all the three components (support data, EWC regularizer and feature regularizer) contribute to the performance of SupportNet to different degrees. Notice that even with only support data, SupportNet can already outperform iCaRL, which shows the effectiveness of our support data selector. The result on CIFAR-100 will be discussed in more detail in Section 4.2. Detailed results about different methods' performance on different classes (confusion matrix) and on the old classes and the new classes separately (accuracy matrix) can be referred to Section B and C in the Appendices. We also show the effectiveness of the consolidation regularizers on stabilizing the learned feature representation in Section D with t-SNE visualization (Maaten & Hinton, 2008) in the Appendices. Furthermore, we compared SupportNet with iCaRL on an additional dataset, tiny ImageNet, which contains 200 classes. The results are shown in Section F in the Appendices, which further demonstrate the effectiveness of SupportNet.

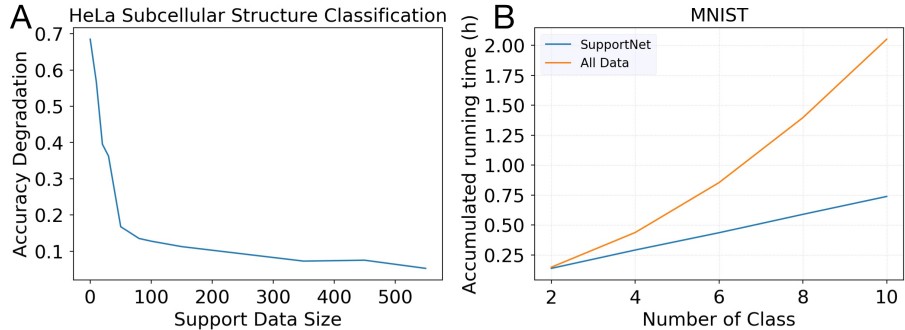

Figure 4: (A): The accuracy deviation of SupportNet from the "All Data" method with respect to the size of the support data. The x-axis shows the support data size. The y-axis is the test accuracy deviation of SupportNet from the "All Data" method after incrementally learning all the classes of the HeLa subcellular structure dataset. (B): The accumulated running time comparison between SupporNet and "All Data" method on MNIST.

Table 1: Performance of SupportNet with respect to different values of the EWC regularizer coefficient. The experiments were done on the enzyme function prediction task. All the results, except for the last two columns, are by incrementally learning all the six classes of the EC system one by one using different EWC regularizer coefficient values, with the support data size fixed to be 2,000. 'SN' stands for SupportNet. The numbers inside the bracket are the coefficient values. The last two columns show the performance of the "All Data" method and iCaRL with the examplar size as 2,000, respectively. The best performance of SupportNet is shown in bold.

| Criteria | SN(1) | SN(10) | SN(100) | SN(1e3) | SN(1e4) | SN(1e5) | All Data | iCaRL |
|---|---|---|---|---|---|---|---|---|
| Accuracy | 0.753 | 0.823 | 0.811 | **0.892** | 0.827 | 0.816 | 0.918 | 0.629 |
| Kappa Score | 0.685 | 0.768 | 0.759 | **0.856** | 0.775 | 0.763 | 0.890 | 0.542 |
| Macro F1 | 0.714 | 0.737 | 0.771 | **0.848** | 0.783 | 0.771 | 0.885 | 0.607 |
| Macro Precision | 0.736 | 0.744 | 0.759 | **0.838** | 0.779 | 0.758 | 0.881 | 0.665 |
| Macro Recall | 0.779 | 0.774 | 0.842 | **0.865** | 0.835 | 0.832 | 0.889 | 0.667 |

## 3.4 SUPPORT DATA SIZE AND RUNNING TIME

As reported by the previous study (Rebuffi et al., 2016), the preserved dataset size can affect the performance of the final model significantly. We investigated that in details here. As shown in Fig. 4 (A), the performance degradation of SupportNet from the "All Data" method decreases gradually as the support data size increases, which is consistent with the previous study using the rehearsal method (Rebuffi et al., 2016). What is interesting is that the performance degradation decreases very quickly at the beginning of the curve, so the performance loss is already very small with a small number of support data. That trend demonstrates the effectiveness of our support data selector, i.e., being able to select a small while representative support dataset. We also show the performance of SupportNet with 2000, 1500, 1000, 500, 200 support data, respectively, in Section E in the Appendices, which further demonstrates the effective of our method. On the other hand, this decent property of our framework is very useful when the users need to trade off the performance with the computational resources and running time. As shown in Fig. 4 (B), on MNIST, SupportNet outperforms the "All Data" method significantly regarding the accumulated running time with only less than 1% performance deviation, trained on the same hardware (GTX 1080 Ti).

## 3.5 REGULARIZER COEFFICIENT

Although the performance of the EWC method on incremental learning is not impressive (Fig. 3), the EWC regularizer plays an important role in our method. Here, we evaluated our method by varying the EWC regularizer coefficient from 1 to 100,000, and compared it with the "All Data" method and iCaRL (Table 1). We can find that the performance of SupportNet varies with different EWC regularier coefficients, with the highest one very close to the "All Data" method, which is

Table 2: Underfitting and overfitting of iCaRL and SupportNet. The experiments were done on the enzyme function prediction data and MNIST. "Real training data" means the training accuracy on the new data plus the support data for SupportNet and examplars for iCaRL. "All training data" means the accuracy of the model trained on the real training data and tested over the new data and all the old data. "Test data" means the accuracy of the model trained on the real training data over the test data.

| Dataset | Enzyme data | | MNIST | |
|---|---|---|---|---|
| Method | SupoortNet | iCaRL | SupoortNet | iCaRL |
| Real training data | 0.987 | 0.991 | 0.998 | 0.995 |
| All training data | 0.920 | 0.626 | 0.991 | 0.902 |
| Test data | 0.839 | 0.629 | 0.988 | 0.878 |

the upper bound of all the incremental learning methods, whereas the lowest one having around 13% performance degradation. The results make sense because from the neurophysiological point of view, SupportNet is trying to reach the stability-plasticity balance point for this classification task. If the coefficient is too small, which means we do not impose enough constraint on those weights which contribute significantly to the old class classification, the deep learning model will be too plastic and the old knowledge tends to be lost. If the coefficient is too large, which means that we impose very strong constraint on those weights even when they are not important to the old class classification, the deep learning model will be too stable and does not have enough capacity to incorporate new knowledge. In general, our results are consistent with the stability-plasticity dilemma.

## 4 DISCUSSION

### 4.1 UNDERFITTING AND OVERFITTING

When training a deep learning model, one can encounter the notorious overfitting issue almost all the time. It is still the case for training an incremental learning model, but we found that there are some unique issues of such learning methods. Table 2 shows the performance of SupportNet and iCaRL on the real training data (i.e., the new data plus the support data for SupportNet and examplars for iCaRL), all the training data (i.e., the new data plus all the old data), and the test data. It can be seen that both methods perform almost perfectly on the real training data, which is as expected. However, the performances of iCaRL on the test data and all the training data are almost the same, both of which are much worse than that on the real training data. This indicates that iCaRL is overfitted to the real training data but underfitted to all the training data. As for SupportNet, the issue is much less severe than iCaRL as the performance degradation from the real training data to all the training data reduces from 37% as in iCaRL to 7% in SupportNet. This suggests that the support data selected by SupportNet are indeed critical for the deep learning training for the old classes. We can find the same pattern on the MNIST dataset.

### 4.2 SUPPORT VECTOR EVOLVING

Despite the impressive performance of SupportNet as shown in Fig. 3, we have to admit the limitation of SupporNet. In fact, using our method, we assume the support vectors of one class will stay static if the learned representation is largely fixed. However, this assumption does not hold under all the circumstances. For example, suppose we perform a binary classification for one very specific type of cat, such as Chartreux, and one very specific type of dog, such as Rottweiler. Later, we need to equip the classifier with the function to recognize another very specific type of cat, such as British Shorthair. We may find that the support vectors of Chartreux change as British Shorthair comes in because Chartreux and British Shorthair are so similar that using the previous support vectors, we are unable to distinguish them. Although SupportNet can still reach the state-of-the-art performance even under this circumstance, as shown in Fig. 3 (F), more work should be done in the future to handle this support vector evolving problem.

## 5 Conclusion

In this paper, we proposed a novel class incremental learning method, SupportNet, to solve the catastrophic forgetting problem by combining the strength of deep learning and SVM. SupportNet can identify the support data from the old data efficiently, which are fed to the deep learning model together with the new data for further training so that the model can review the essential information of the old data when learning the new information. With the help of two powerful consolidation regularizers, the support data can effectively help the deep learning model prevent the catastrophic forgetting issue, eliminate the necessity of retraining the model from scratch, and maintain stable learned representation between the old and the new data.

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

# Appendices

## A   DERIVATION OF EQUATION 3 FROM EQUATION 2

In the section, we use chain rule to derive the following equation

$$-\frac{\partial L}{\partial w_{j,i}} = \frac{1}{N} \sum_{n=1}^{N} (\tilde{y}_{n,i} - o_{n,i}) \delta_{n,j}, \tag{12}$$

from

$$L = -\frac{1}{N} \sum_{n=1}^{N} \sum_{k=1}^{K} \tilde{y}_{n,k} \log(o_{n,k}). \tag{13}$$

Let us first consider just one data sample:

$$L_n = -\sum_{k=1}^{K} \tilde{y}_{n,k} \log(o_{n,k}). \tag{14}$$

Using chain rule, we have

$$-\frac{\partial L_n}{\partial w_{j,i}} = -\sum_{l=1}^{K} \frac{\partial L_n}{\partial o_{n,l}} \frac{\partial o_{n,l}}{\partial z_{n,i}} \frac{\partial z_{n,i}}{\partial w_{j,i}}, \tag{15}$$

For the first term in Eq. 15, we have

$$\frac{\partial L_n}{\partial o_{n,l}} = \frac{\partial - \sum_{k=1}^{K} \tilde{y}_{n,k} \log(o_{n,k})}{\partial o_{n,l}}$$
$$= -\frac{\tilde{y}_{n,l}}{o_{n,l}}. \tag{16}$$

For the second term in Eq. 15, we have

$$\frac{\partial o_{n,l}}{\partial z_{n,i}} = \frac{\partial \frac{\exp(z_{n,l})}{\sum_{k=1}^{K} \exp(z_{n,k})}}{\partial z_{n,i}}$$
$$= \frac{\frac{\partial \exp(z_{n,l})}{\partial z_{n,i}} \sum_{k=1}^{K} \exp(z_{n,k}) - \exp(z_{n,l}) \exp(z_{n,i})}{(\sum_{k=1}^{K} \exp(z_{n,k}))^2} \tag{17}$$
$$= \begin{cases} o_{n,i}(1 - o_{n,i}), & l = i \\ -o_{n,i} o_{n,l}, & l \neq i. \end{cases}$$

For the third term in Eq. 15, we have

$$\frac{\partial z_{n,i}}{\partial w_{j,i}} = \frac{\partial \mathbf{W}_{i,:} \boldsymbol{\delta}_n}{\partial w_{j,i}}$$
$$= \delta_{n,j}. \tag{18}$$

Put Eq. 16, Eq. 17, and Eq. 18 into Eq. 15, we have:

$$-\frac{\partial L_n}{\partial w_{j,i}} = (\frac{\tilde{y}_{n,i}}{o_{n,i}} o_{n,i}(1 - o_{n,i}) + \sum_{l \neq i}^{K} \frac{\tilde{y}_{n,l}}{o_{n,l}} (-o_{n,i} o_{n,l})) \delta_{n,j}$$
$$= (\tilde{y}_{n,i} - o_{n,i} \sum_{l=1}^{K} \tilde{y}_{n,l}) \delta_{n,j} \tag{19}$$
$$\overset{(1)}{=} (\tilde{y}_{n,i} - o_{n,i}) \delta_{n,j},$$

where (1) is the result of the fact that we use one hot encoding for the label and $\sum_{l=1}^{K} \tilde{y}_{n,l} = 1$.

From Eq. 19, we can easily get Eq. 12 by considering all the data points.

# B CONFUSION MATRICES

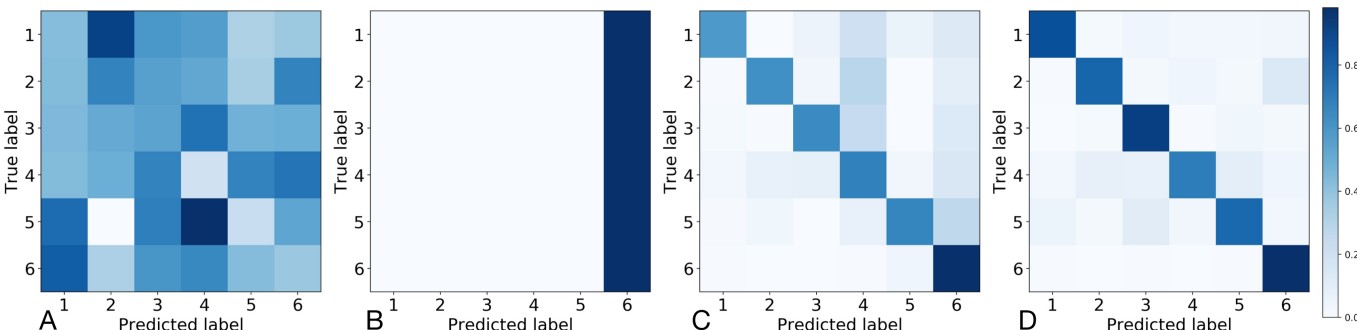

Figure 5: The confusion matrix of different methods on the 6-class classification task for EC prediction: (A) the "Random Guess" method, (B) the "Fine Tune" method, (C) iCaRL, and (D) Support-Net. The data from the first five classes were given as the old data, and the ones from the sixth class were given as the new data.

We investigate the confusion matrices of the "Random Guess" method, the "Fine Tune" method, iCaRL and SupportNet (Fig. 5) after the last batch of classes on the EC data. As expected, the "Fine Tune" method only considers the new data from the new class, and thus is overfitted to the new class (Fig. 5(B)). The iCaRL method partially solves this issue by combining deep learning with nearest-mean-examplars, which is a variant of KNN (Fig 5(C)). SupportNet, on the other hand, combines the advantage of SVM and deep learning by using SVM to find the important support data, which efficiently preserve the knowledge of the old data, and utilizing deep learning as the final classifier. This novel combination can efficiently and effectively solve the incremental learning problem (Fig 5(D)). Notice that the upper left diagonal of the SupportNet's confusion matrix has much higher values than those of the iCaRL's confusion matrix, which indicates the performance improvement comes from the accuracy prediction of the old classes.

# C ACCURACY MATRICES

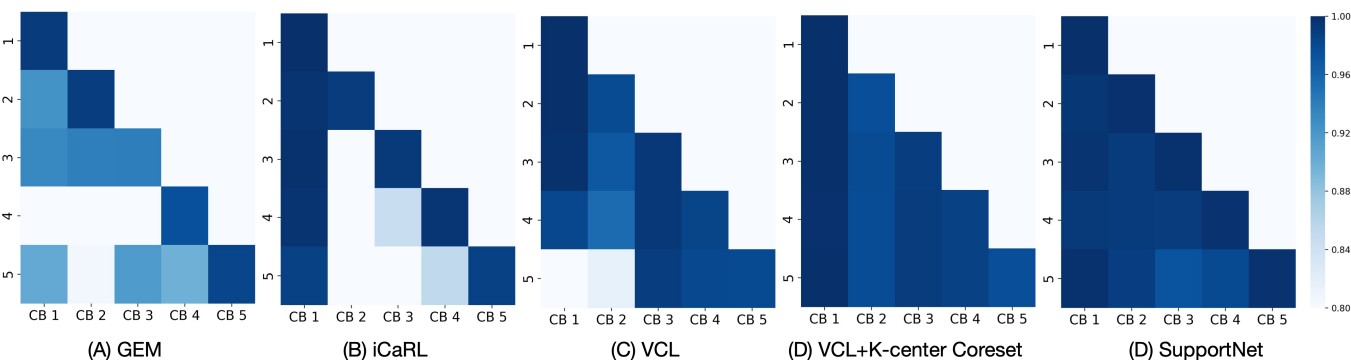

Figure 6: The accuracy matrices of different methods on MNIST. Those matrices show the performance composition of Fig 3 (A), considering those methods' performance on the classes belonging to different class batch (CB) separately. In the matrix, each row represents the performance of the deep learning model at each incremental training time point. Each column represents the performance of the deep learning model on each class batch's test data. (A) GEM's accuracy matrix on MNIST. (B) iCaRL's accuracy matrix on MNIST. (C) VCL's accuracy matrix on MNIST. (D) VCL with K-center Coreset's accuracy matrix on MNIST. (E) SupportNet's accuracy matrix on MNIST.

In this section, we investigate the performance composition of SupportNet on MNIST shown in Fig. 3 (A). Fig. 3 (A) only shows the overall performance of different methods on all the testing data,

averaging the performances on the old test data and the new test data, which can lose the insight of different methods' performance on old data. To avoid that, we further check the performance of different methods on the old data and the new data separately, whose results can be referred to Fig. 6. As shown in Fig. 6 (B), iCaRL can maintain its performance on the oldest class batch very well, however, it is unable to maintain its performance on the intermediate class batches. GEM (Fig. 6 (A)) can outperform iCaRL on the middle class batches, however, it cannot maintain the performance of the oldest class batch. VCL (Fig. 6 (C)) further outperforms GEM in terms of middle class batches, however it suffers from the same problem as GEM, being unable to preserve the performance on the oldest class batch. On the other hand, both VCL with K-center Coreset and SupportNet can maintain their performance on the old data classes almost perfectly, no matter for the intermediate class batches or the oldest class batch. However, because of the difference between the two algorithms, their trade-offs are different. Although VCL with K-center Coreset can maintain the performance of old classes almost exactly, there is a trade-off of the methods on the newest classes, with the newest model being unable to achieve the optimal performance on the newest class. As for SupportNet, it allows slight performance degradation on the old classes while can achieve optimal performance on the newest class batch.

## D  T-SNE VISUALIZATION OF FEATURE REPRESENTATION

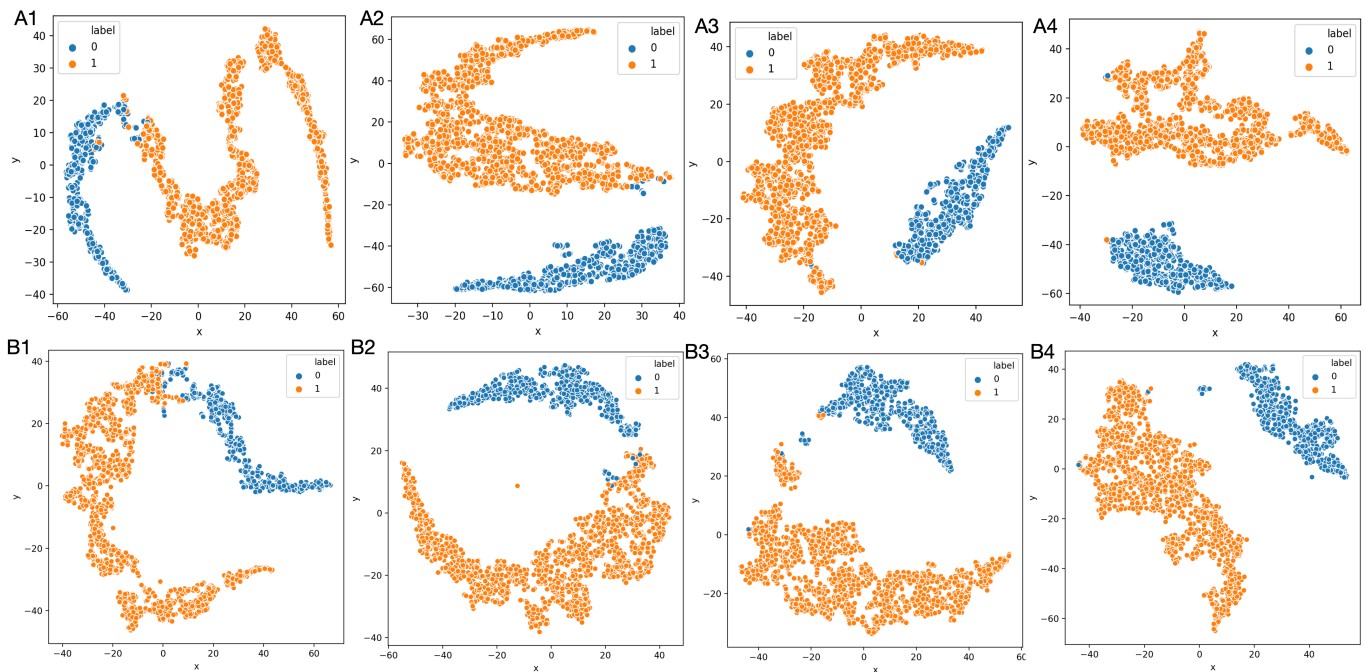

Figure 7: The t-SNE visualization of EC dataset's feature representation at each incremental training time point. (A1-A4). t-SNE result of the feature representation learned from SupportNet without any regularizers. (B1-B4). t-SNE result of the feature representation learned from SupportNet with the consolidation regularizers.

The feature representation learned by the deep learning models during the incremental learning process is worth investigating, since it can suggest why SupportNet works to a certain degree. We take the EC dataset and the MNIST dataset as examples and use t-SNE (Maaten & Hinton, 2008) to investigate the learned representation. For each dataset, we randomly select 2000 data points from the training data at the first training time point. Then, after each future training time point, we apply the further trained model to the selected data points and extract the input of the deep learning model's last layer as the learned feature representation. After obtaining those raw feature representations, we apply t-SNE to them and visualize them in 2D space. For each dataset, we investigated both the SupportNet with consolidation regularizers and SupportNet without any regularizers. The result of EC data can be referred to Fig. 7 and the result of MNIST data can be referred to Fig. 8. As

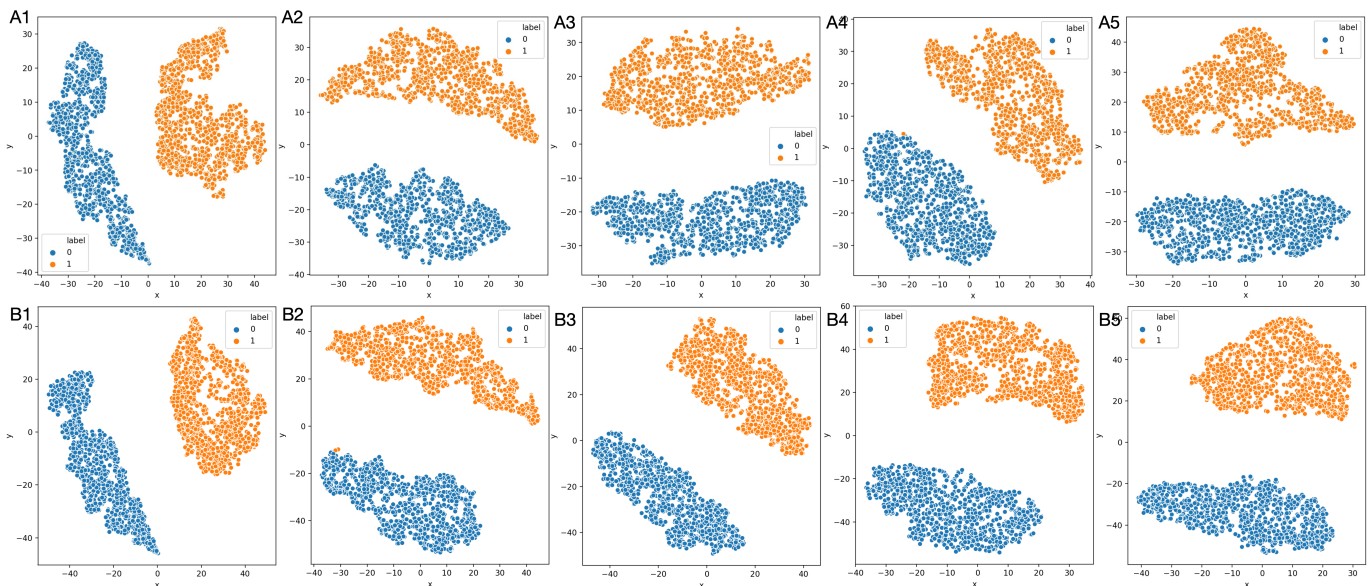

Figure 8: The t-SNE visualization of MNIST dataset's feature representation at each incremental training time point. (A1-A5). t-SNE result of the feature representation learned from SupportNet without any regularizers. (B1-B5). t-SNE result of the feature representation learned from Support-Net with consolidation regularizers.

shown in those figures, although the feature representation of the standard SupportNet still varies, compared to the SupportNet without any regularizers, the variance is much smaller, which suggests that the consolidation regularizes help the model stabilize the learned feature representation.

## E    PERFORMANCE ON MNIST WITH LESS SUPPORT DATA

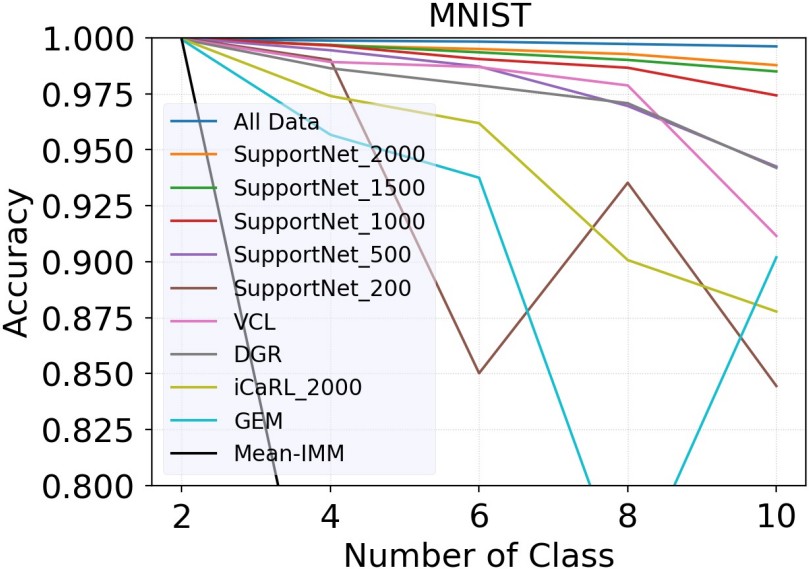

Figure 9: Performance of SupportNet with less support data. The experiment setting is the same as Fig. 3, except for that we use less support data. 'SupportNet_500' means that we use only 500 data points as support data.

In this section, we further investigate the performance of SupportNet with less support data as a supplement of Section 3.4. We run the experiments of SupportNet with the support data size as 2000, 1500, 1000, 500, and 200, respectively, whose results are shown in Fig. 9. As shown in the figure, even SupportNet with 500 support data points can outperform iCaRL with 2000 examplars, which further demonstrates the effectiveness of our support data selecting strategy .

## F  PERFORMANCE ON TINY IMAGENET

To further evaluate SupportNet's performance on class incremental learning setting with more classes, we tested it on tiny ImageNet dataset[4], comparing it with iCaRL. The setting of tiny ImageNet dataset is similar to that of ImageNet. However, its data size is much smaller than ImageNet. Tiny ImageNet has 200 classes while each class only has 500 training images and 50 testing images, which means that it is even harder than ImageNet. The performance of SupportNet and iCaRL on this dataset is shown in Fig. 10. As illustrated in the figure, SupportNet can outperform iCaRL significantly on this dataset. Furthermore, as suggested by the red line, which shows the performance difference between SupportNet and iCaRL, SupportNet's performance superiority is increasingly significant as the class incremental learning setting goes further. This phenomenon demonstrates the effectiveness of SupportNet in combating catastrophic forgetting.

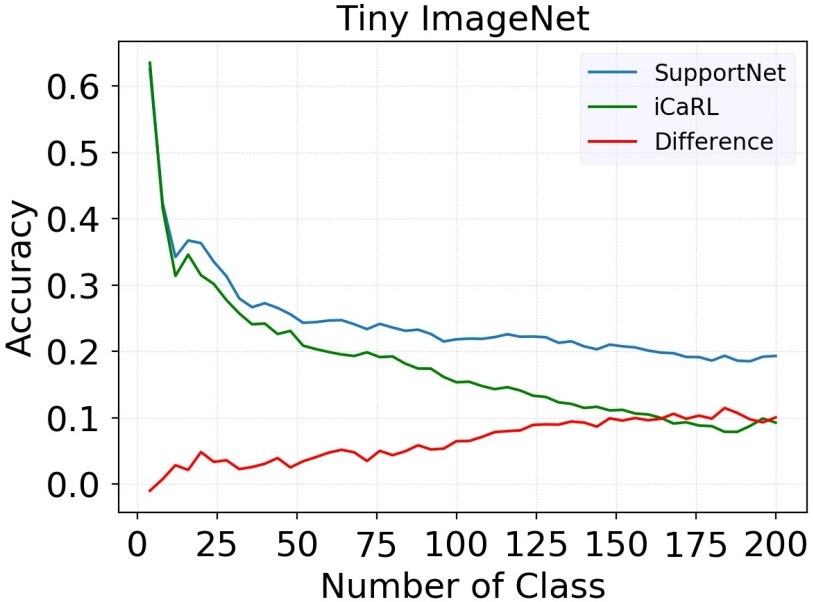

Figure 10: Performance comparison of SupportNet and iCaRL on tiny ImageNet dataset. The experiment setting is the same as Fig. 3, except for that we have more classes. 'Difference' shows the performance difference between SupportNet and iCaRL along the training process.

---

[4]https://tiny-imagenet.herokuapp.com/

