# OpenReview forum: "SupportNet: solving catastrophic forgetting in class incremental learning with support data"
_ICLR.cc/2019/Conference_

### Official Review · AnonReviewer2 · 2018-11-02
**Redundant idea**

**Rating:** 4
**Confidence:** 4

**Review:**

This paper presents a continual learning method that aims to overcome the catastrophic forgetting problem by holding out small number of samples for each task to be used in training for new tasks. Specifially, these representative samples for each task are selected as support vectors of a SVM trained on it. The proposed method, SupportNet, is validated on a continual learning task of a classifier against two existing continual learning approaches, which it outperforms.

Pros
- Idea of using SVM to identify the most important samples for classification makes sense.

Cons
- The idea of storing a small subset of a original dataset for each task has been already explored in [Nguyen et al. 18], and thus is not novel.
- Thus the contribution of this work reduces to the use of SVM to identify the most important samples, but the effectiveness of this approach is not validated since it does not compare against [Nguyen et al. 18].
- Also it leaves out many of the recent work on continual learning.
- The idea of using SVM for identifying important samples is not very attractive since an SVM will have a very different decision boundary from the one from the trained DNN.
- Also this method is only applicable to a classification task and not to other tasks such as regression or RL.

Thus considering the lack of novelty and experimental validation, I recommend rejecting this paper.

[Nguyen et al. 18] Variational Continual Learning, ICLR 2018

---

> ### Author Response · Authors · 2018-11-16
> **Response to AnonReviewer2**
>
> We would like to thank the reviewer for the comments. Please find below for the detailed responses to each comment.
>
> ------------------------------------
> (1,2)
> "
> - The idea of storing a small subset of a original dataset for each task has been already explored in [Nguyen et al. 18], and thus is not novel.
> - Thus the contribution of this work reduces to the use of SVM to identify the most important samples, but the effectiveness of this approach is not validated since it does not compare against [Nguyen et al. 18].
> "
>
> Reply: Thank you very much for pointing out this very recent paper [Nguyen et al. 18], which was published several months ago. The method is indeed a very good method and we are aware of it months ago. Unfortunately, at that time, we have already finished the manuscript and did not compare with it in the original submission. In the revision, we added the comparison of SupportNet with the three versions of this method and all the other methods pointed out by the other reviewer. We performed the experiments on the dataset that was used by all the methods, MNIST. We used the code published by the authors and modified it to make the experiment environment being consistent. As suggested by Fig. 3 (A), this very recent method, VCL, indeed performed very impressively, outperforming all the other method, except for SupportNet. In fact, the coreset version of VCL's performance is very close to that of SupporNet. But nevertheless, SupportNet can still outperform all the three versions of VCL along the incremental learning process.
>
> However, we would like to further clarify on your comment, which, we believe, is somehow unfair. First, [Nguyen et al. 18] was published after we almost finished our initial paper. Second, using the subset of old data for further training is indeed not a novel idea, but it is a category of methods, called rehearsal methods [Parisi et al. 18]. It was proposed long before [Nguyen et al. 18]. iCaRL [Rebuff et al. 17] and GEM [Lopez-Paz and Ranzato, 2017] also belong to this category. Third, not comparing against [Nguyen et al. 18] does not mean we were unable to or did not validate the effectiveness of the proposed method in the original submission. In fact, we compared our method with the iCaRL [Rebuff et al. 17] in the original submission, which is a representative and previous state-of-the-art method within this category, as suggested by [Kemker et al. 18]. Furthermore, we compared our method with the performance empirical upper bound method: training new models from scratch using all the available dataset at each time point. All the results in the original submission clearly suggested the effectiveness of using SVM to identify the most important samples since it can outperform iCaRL with a large margin on all the dataset and even achieve near-optimal performance on EC dataset and MNIST. Fourth, proposing a simple and effective method of selecting the important data point is not a marginal contribution. As suggested by [Parisi et al. 18], for this kind of methods, the most important things are to find the most useful and important points to preserve and to build up the framework to take advantage of the old data and the new data. Furthermore, instead of being just the support data selector, SupportNet is a framework, which contains both the data selector and the consolidation regularizers. The effectiveness of its components is well-tested by the comprehensive experiments in the manuscript.
>
> ------------------------------------
> (3) "Also it leaves out many of the recent work on continual learning."
>
> Reply: Thank you very much for raising this concern! In the original version, we compared our method with the previous state-of-the-art method for class incremental learning, iCaRL, as suggested by [Kemker et al. 18]. However, your concern is reasonable. We have added the comparison of SupportNet with all the other recent methods that have been pointed out by the reviewers, which is shown in Fig. 3 (A). The comprehensive experiments demonstrate the effectiveness of SupportNet.
>
>
> ---Thank you again and we will continue in the next post---

---

> > ### Author Response · Authors · 2018-11-16
> > **Response to AnonReviewer2--continued**
> >
> > (4) "The idea of using SVM for identifying important samples is not very attractive since an SVM will have a very different decision boundary from the one from the trained DNN."
> >
> > Reply: Thank you very much for mentioning this. Yes, you are right about the difference between the decision boundary of deep neural network and that of SVM. However, that is not the point of our paper.
> >
> > In fact, the decision boundary of the whole neural network can be very different from that of SVM which is trained from the original data. However, what we discussed in the manuscript is the decision boundary of the neural network's last layer and the SVM trained from the input of the network's last layer.
> > To be more specific, we can consider all the layers in the neural network before the last layer as a feature representation learner and the last layer as the classifier. The feature representation learner can transfer the original data into another space, where the classifier performs the classification. If we train a hard margin SVM within that transferred space, the decision boundary of that SVM can be very similar to that of the classifier in the deep learning model. Under some assumptions, they can be proved to be the same.
> > As we referred in the manuscript, you can refer to [Soudry et al. 18] for the theoretical analysis. In terms of thorough experiments, you can refer to [Li et al. 18]. Although the assumption is strong (separable data) to prove the decision boundary of SVM is the same as the deep learning's last layer, in practice, the two decision boundaries are very similar as shown in [Li et al. 18].
> >
> > Moreover, as we discussed in Section 2.1, after the learned representation becoming stable, as we further train the neural network, it is the support data that will contribute largely to the loss and the gradient, which makes the combination of SVM and deep learning to deal with catastrophic forgetting a natural thing.
> >
> > ------------------------------------
> > (5) "Also this method is only applicable to a classification task and not to other tasks such as regression or RL."
> >
> > Reply: In this manuscript, we focused on the class incremental learning, which is one of the main tasks of continual learning as suggested by [Kemker et al. 18] and can have broad applications in various fields, such as computer vision and biomedical informatics. However, the idea of combining SVM and deep learning can be generalized to other continual learning tasks, although more efforts need to be done.
> >
> > ------------------------------------
> > Thank you again for reviewing our paper. Please take a look at the updated version and please let us know if you have more comments.
> >
> >
> > [Lopez-Paz and Ranzato, 2017] Gradient Episodic Memory for Continual Learning, NIPS 2017
> > [Parisi et al. 18] Continual Lifelong Learning with Neural Networks: A Review, arxiv.org/abs/1802.07569
> > [Kemker et al. 18] Measuring Catastrophic Forgetting in Neural Networks, AAAI 2018
> > [Rebuff et al. 17] iCaRL: Incremental Classifier and Representation Learning, CVPR 2017
> > [Nguyen et al. 18] Variational Continual Learning, ICLR 2018

---

> > ### Comment · AnonReviewer2 · 2018-12-07
> > **Reply to author response.**
> >
> > Thank you for providing the additional experimental results with VCL. However, it is difficult to justify that the proposed method is superior to coreset method based on the results from the MNIST experiments alone. Why didn't you report the results on other two datasets? How does SupportNet fare against VCL on the other two datasets?
> >
> > Also, since VCL is an ICLR 2018 paper and was already out for more than 5 months at the time of ICLR 2019 submission, not being aware of it at its initial submission does not sound like a good execuse.

---

> > > ### Author Response · Authors · 2018-12-07
> > > **Thank you very much for your further comments!**
> > >
> > > Thank you very much for your further reply!! We are glad to hear your further comments.
> > >
> > > ##### for the further comparison dataset #####
> > > As we mentioned in the previous rebuttal, we want a fair comparison between all the additional methods pointed outed by the reviewers. So, we chose MNIST, which is the intersection dataset between all the methods, as the evaluation dataset. Furthermore, in the VCL paper [Nguyen et al. 18], although there are three experiments, 'Permuted MNIST', 'Split MNIST', 'Split notMNIST', they are all related to MNIST. We just followed the dataset that was used by VCL on this class incremental learning problem.
> > >
> > > ##### for the performance #####
> > > We admit the nice performance of VCL, however, as shown in Figure 3 (A), SupportNet's performance is higher than all the three version of VCL along the incremental learning process on the dataset.
> > >
> > > On the other hand, we should admit every method has their own limitations and assumptions, including both VCL and SupportNet. No method will be superior to the other methods under all the circumstances and all the datasets. For example, even the deep learning method itself can fail under certain settings [Shai et al. 17]. In terms of SupportNet, as we discussed in Section 4.2, SupportNet will encounter the 'Support Vector Evolving' problem if the new dataset is highly related to the old dataset. If we break the assumption and design a specific dataset which attacks SupportNet, SupportNet can have inferior performance. Nevertheless, it works fine on real datasets, as shown in the experiments.
> > >
> > > Preparing this paper, we do not mean to outperform all the methods under all the circumstances. We hope the idea conveyed by our manuscript can be useful or inspirational for people to finally overcome catastrophic forgetting. We are also glad to know there is another paper submitted to ICLR2019 sharing related idea [Anonymous 19].
> > >
> > > ##### for missing VCL in the original version of SupportNet #####
> > > Thank you very much for your further comments about missing VCL in the original submission! We are very sorry for not mentioning VCL in the original submission. As we mentioned in the previous rebuttal, we are aware of it months ago. Unfortunately, at that time, we have already finished the manuscript and did not compare with it in the original submission. Basically, we finished the first draft in April, whose records are traceable on the web. But please do not do so until later, otherwise, it will break the double-anonymous rule.
> > >
> > > Nevertheless, we have referred VCL in the revision for reader's reference and added the comparison with VCL as you commented.
> > >
> > > #########################
> > > Thank you very much for your comments about the performance and the comparison between SupporNet and VCL, which definitely improves the quality of our manuscript. We hope we could hear more comments from you regarding other aspects of our paper, which could further improve the quality of our method and manuscript.
> > >
> > >
> > > [Shai et al. 17] Failures of Gradient-Based Deep Learning, ICML 2017
> > > [Nguyen et al. 18] Variational Continual Learning, ICLR 2018
> > > [Anonymous 19] An Empirical Study of Example Forgetting during Deep Neural Network Learning, submitted to ICLR 2019

---

### Official Review · AnonReviewer1 · 2018-11-02
**Interesting idea but limited comparisons**

**Rating:** 5
**Confidence:** 4

**Review:**

This paper presents a hybrid concept of deep neural network and support vector machine (SVM) for preventing catastrophic forgetting. The authors consider the last layer and the softmax function as SVM, and obtain support vectors, which are used as important samples of the old dataset. Merging the support vector data and new data, the network can keep the knowledge on the previous task. The use of support vector concept is interesting, but this paper has some issues to be improved.

Pros and Cons
  (+) Interesting idea
  (+) Diverse experimental results on six datasets including benchmark and real-world datasets
  (-) Lack of related work on recent catastrophic forgetting
  (-) Limited comparing results
  (-) Limited analysis of feature regularizers

Detailed comments
- I am curious how we can assure that SVM's decision boundary is similar or same to NN's boundary
- SupportNet is a method to use some of the previous data. For fair comparisons, SupportNet needs to be compared with other models using previous samples such as GEM [Lopez-Paz and Ranzato, 2017].
- Following papers are omitted in related work:
  1. Lee et al. Overcoming Catastrophic Forgetting by Incremental Moment Matching, NIPS 2017.
  2. Shin et al. Continual Learning with Deep Generative Replay, NIPS 2017.
   Also, the model needs to be compared with two models.
- There is no result and analysis for feature regularizers. As the authors referred, the features of support vector data continuously change as the learning goes on. So, I am curious how the feature regularizer has effects on the performance.  This can be performed by visualizing the change of support vector features via t-SNE as the incremental learning proceeds
- The authors used 2000 support vectors for MNIST, Cifar-10, and Cifar-100. However, this size might be quite large considering their difficulty.
- How is the pattern of EwC using some samples in the old dataset?
- iCaRL was evaluated on ImageNet. Is there any reason not to be evaluated on ImageNet?
- What kind of NNs is used for each dataset? And what kind of kernel is used for SVM?

---

> ### Author Response · Authors · 2018-11-16
> **Response to AnonReviewer1**
>
> For reviewer 1:
> We would like to thank the reviewer for the detailed comments, which helped us improve the manuscript. Please find below the detailed response to each comment. If possible, could you please explain more on comment (6)? We did not fully understand the question.
>
> Detailed comments
> ------------------------------------
> (1) "I am curious how we can assure that SVM's decision boundary is similar or same to NN's boundary"
>
> Reply: Thank you for asking this, which is the basis of SupporNet. In fact, the decision boundary of the whole neural network can be very different from that of SVM trained from the original data. However, what we discussed in the manuscript is the decision boundaries of the neural network's last layer and that of SVM trained from the input of the network's last layer. For more details, as we referred in the manuscript, please refer to [Soudry et al. 18] for the theoretical analysis. In terms of thorough experiments, you can refer to [Li et al. 18]. Although the assumption needs to be strong (separable data) to prove the decision boundary of SVM is the same as that of the deep learning's last layer, in practice, the two decision boundaries are very similar as shown in [Li et al. 18].
>
> ------------------------------------
> (2) "SupportNet is a method to use some of the previous data. For fair comparisons, SupportNet needs to be compared with other models using previous samples such as GEM [Lopez-Paz and Ranzato, 2017]."
>
> Reply: Thank you for pointing out this related work. In our experiments, we did compare our method with another method using the sampled old data idea, iCaRL, which is suggested by [Kemker et al. 18] to be the previous state-of-the-art method for class incremental learning. But following your suggestion, we added the comparison of SupportNet with GEM [Lopez-Paz and Ranzato, 2017] and the other two methods you mentioned in the revision, using MNIST, which is the shared dataset of all the methods. We used the code released by the authors, except for modifying the code to make the experiment setting being consistent with the other methods. As suggested by Figure 3 (A), all the three methods suggested by you indeed performed well, however, our method can still outperform the three methods with a large margin.
>
> ------------------------------------
> (3) "Following papers are omitted in related work:..."
>
> Reply: Thank you very much for pointing out these two interesting works. We added the comparison with the two methods, whose results are shown in Figure 3 (A). As suggested by the figure, these two methods indeed perform relatively well, however, SupportNet can still clearly outperform the two methods.
>
> ------------------------------------
> (4) "There is no result and analysis for feature regularizers. As the authors referred, the features of support vector data continuously change as the learning goes on. So, I am curious how the feature regularizer has effects on the performance.  This can be performed by visualizing the change of support vector features via t-SNE as the incremental learning proceeds"
>
> Reply: Thank you very much for raising this up. Feature regularizer analysis is indeed important, which is also suggested by reviewer 3. Following the reviewer 3's suggestion, we have added the performance comparison of SupportNet only using feature regularizer and only using support data, whose results are shown in Fig.3 (C) and which shows that the feature regularizer indeed contributes to the performance gain of SupportNet. But your suggestion is also very constructive. Using t-SNE can help us understand what the feature representation has been learned by the model. However, only checking the feature representation of the support data is not enough. In fact, we can constrain the feature representation of the support data very strictly by using a large feature regularizer coefficient. This will have two disadvantages. First, the model's flexibility is reduced and the model has less capacity to learn new classes. Second, the feature representation of non-support data may change which can have a negative impact on the model's performance. In fact, what we want is the stability of the feature representation of all the old data instead of just the support data. To show that, we added the t-SNE plotting of feature representation of 2000 random data points along the training process in Section D in the Appendix. As shown in the figures, the feature representation still varies, but compared with those which we do not apply regularizers, the shape variance is much smaller. This suggests that the EWC regularizer and feature regularizer can indeed stabilize the hidden representation of old data, which is important for the effectiveness of SupportNet.
>
>
> ---Thank you again and we will continue in the next post---

---

> > ### Author Response · Authors · 2018-11-16
> > **Response to AnonReviewer1--continued**
> >
> > (5) "The authors used 2000 support vectors for MNIST, Cifar-10, and Cifar-100. However, this size might be quite large considering their difficulty."
> >
> > Reply: Thank you very much for asking this. Since we compared SupportNet with iCaRL [Rebuff et al. 17], we wanted the experiment setting to be the same as the iCaRL paper. As a result, We followed the setting in iCaRL [Rebuff et al. 17] and set the support data size to be 2000. On the other hand, we also explored the performance of SupportNet with the variance of support data size, whose result can be found in Fig. 4 (A) and Section 3.4. As shown in the curve of Fig. 4 (A), the performance of SupportNet is very high even when we just use a very small number of support data, which shows the effectiveness of our support data selector. But your concern is reasonable, we further added additional experiments using much fewer support data (1500, 1000, 500, 200) on MNIST, whose result is shown in Section E in the Appendix. As shown in Fig. 9, SupportNet with only 500 support data can already outperform iCaRL with 2000 examplars significantly, which shows the effectiveness of SupportNet.
> >
> > ------------------------------------
> > (6) "How is the pattern of EwC using some samples in the old dataset?"
> >
> > Reply: What do you mean by the pattern of EWC? Do you mean the performance or feature representation? We did not fully understand your question, could you clarify?
> >
> > ------------------------------------
> > (7) "iCaRL was evaluated on ImageNet. Is there any reason not to be evaluated on ImageNet?"
> >
> > Reply: Thank you for asking this. When we designed the experiments, we considered the variance of dataset properties and domain knowledge to ensure the broad usage of our method. As a result, in addition to the two commonly used benchmark datasets in computer vision filed (MNIST and CIFAR-10/100), we also covered some real datasets from other domains and fields, including breast cancer dataset, cancer subcellular structure dataset and enzyme classification dataset. Imagenet is truly a very good dataset, however, it is a little bit redundant considering we have already included MNIST and CIFAR-10/100. We want our method to have broad applications instead of just for the computer vision field. But your concern is reasonable, we run the experiment on tiny ImageNet, which is similar to but even more difficult than ImageNet, regarding the ratio between the data size and the number of classes. Tiny ImageNet has 200 classes while each class only has 500 training images and 50 testing images. The result is shown in Section F and Figure 10. As shown in the figure, SupportNet can outperform iCaRL significantly on this dataset. Furthermore, SupportNet's performance superiority is increasingly significant as the class incremental learning setting goes further, which, again, demonstrates the effectiveness of SupportNet in combating catastrophic forgetting.
> >
> > ------------------------------------
> > (8) "What kind of NNs is used for each dataset? And what kind of kernel is used for SVM?"
> >
> > Reply: Thank you very much for raising this up, which is important for the experiments. As we mentioned in Section 3.2, for the EC data, we used the architecture from [Li et al. 17] and for the other dataset, we used residual network with 32 layers. In terms of the kernel, since the last layer of the deep learning model is basically a linear classifier and as suggested by [Soudry et al. 2018], we used the linear kernel. We further emphasized that in the revision following your question.
> >
> > ------------------------------------
> > Thank you again for providing thoughtful comments to our paper. Please take a look at the updated version and please let us know if you have more comments.
> >
> >
> > [Soudry et al. 18] The Implicit Bias of Gradient Descent on Separable Data, ICLR 2018
> > [Li et al. 18] On the Decision Boundary of Deep Neural Networks, arxiv.org/abs/1808.05385
> > [Kemker et al. 18] Measuring Catastrophic Forgetting in Neural Networks, AAAI 2018
> > [Rebuff et al. 17] iCaRL: Incremental Classifier and Representation Learning, CVPR 2017
> > [Li et al. 17] DEEPre: sequence-based enzyme EC number prediction by deep learning, Bioinformatics 2017

---

### Official Review · AnonReviewer3 · 2018-11-04
**SupportNet offers a new method to perform class incremental learning with an support vectors from the last layers and subsequent two learning constraints, showing some improved performance compared to previous approaches.**

**Rating:** 6
**Confidence:** 4

**Review:**

Summary:
The authors offer a novel incremental learning method called SupportNet to combat catastrophic forgetting that can be seen in standard deep learning models. Catastrophic forgetting is the phenomenon where the networks don’t retain old knowledge when they learn new knowledge.  SupportNet uses resnet network with 32 layers, trains an SVM on the last layer and the support vector points from this SVM are given to the network along with the new data. Furthermore, two regularizers, feature and EWC regularizer, are added to the network. The feature regularizer forces the network to produce fixed representation for the old data, since if the feature representation for the old data changes when the network is fine-tuned on the new data then the support vectors generated from the old feature representation of the old data would become invalid.  The EWC regularizer works by constraining parameters crucial for the classification of the old data, making it harder for the network to change them. SupportNet is compared to five methods (all data: network is re-trained with new and old data, upper bound for performance, iCaRL: state-of-the-art method for incremental learning, EWC: Only EWC regularizer added, Fine tune: Only new data, Random guessing: Random guess to assign labels) on six datasets (MNIST, CIFAR-10, CIFAR-100, Enzyme Function Prediction, HeLa Subcellular Structure Classification, Breast Tumor Classification). It shows some improvement in overall accuracy with each newly added class when compared to iCaRL, EWC, Fine Tune and Random guessing.  Additionally, they show that overfitting for the real training data (a chosen subset of old data and the new data) is a problem for the competition iCaRL and affects SupportNet to a much lesser degree.

Pros:
(1)
The authors propose a sensible approach, which is also novel to be best of our knowledge, using SVM to select support data from old data to be fed to the network along with the new data in the incremental learning framework to avoid catastrophic forgetting. Additionally, they offer a feature regularizer that penalizes the network for changing the feature representation of the support data when training the network on new data and an EWC regularizer that constrains the parameters that are crucial for the classification of the old data and makes it harder to change them.
(2)
The authors use six different datasets and several other approaches (subsets of their method’s components, other competing methods) to show these three components alleviate catastrophic forgetting and show improvement in overall accuracy.
(3)
The paper is well written and easy to follow.


Cons:
Major Points:
(1)
To show that the method proposed in the paper addresses catastrophic forgetting, in addition to the overall accuracy shown in Figure 3, it is also necessary to show the accuracy of different models on old classes when new classes are added to the network. This will strengthen the argument that the improvement in accuracy is indeed due to correct classification on old data.
 (2)
The authors claim that iCaRL suffers from overfitting on real training data (section 4.1) however Table 2 shows iCaRL only on the enzyme function prediction which is also the dataset where the difference in performance between iCaRL and SupportNet is the largest. To support the general overfitting claim made in section 4.1, the authors should repeat this analysis on any of the other five datasets where the performance difference between the two methods is much smaller.  SupportNet also suffers from overfitting (Table 3, Accuracy: test data: 83.9%, real training data: 98.7%) although to a lesser extent than iCaRL.
(3)
The individual impact of the support points and the joint impact of support points with feature regularizer on accuracy is not assessed. To prove their usefulness, add two methods to Figure 3:
(a)A method that uses support points without any regularizer.
(b) A method that uses support points with just the feature regularizer.

Other points:
(1)
In section 2.3.2, EWC regularizer, Eq. 9: We think F(theta_new) should be F(theta_old) since we want to constrain parameters crucial for classification of old data and should be computing Fisher Information for the old parameters.
(2)
In section 2.1 Deep Learning and SVM: additional steps are needed to show how Eq. 3 is derived from Eq. 2.
(3)
In section 2.1 Deep Learning and SVM: In the line before Eq. 4. “t represtent” instead of “t represents”.
(4)
Figures are small and hard to read. Please increase the size and resolution of the figures.

---

> ### Author Response · Authors · 2018-11-16
> **Response to AnonReviewer3**
>
> We would like to sincerely thank you for your insightful and detailed comments, which summarized the pros and cons of our manuscript comprehensively and pointed out the direction for us to improve the manuscript. Please find below our point-by-point reply to your comments.
>
>
> Major Points:
> ------------------------------------
> (1). "To show that the method proposed in the paper addresses catastrophic forgetting, in addition to the overall accuracy shown in Figure 3, it is also necessary to show the accuracy of different models on old classes when new classes are added to the network. This will strengthen the argument that the improvement in accuracy is indeed due to correct classification on old data."
>
> Reply: This is a very good suggestions. We initially included a series of confusion matrices in the manuscript to show different methods' performance on the old classes. But since we want to control the number of pages to be around 8, we removed those confusion matrices in the submitted version. We now put the confusion matrices back in Section B in the Appendix. In addition, we added a series of accuracy matrices, showing the performance of different methods on old classes on MNIST along the incremental training process in Section C in the Appendix. Those confusion matrices and accuracy matrices show that SupportNet's performance over the old classes is very close to that of the newest classes and SupportNet's improvement over accuracy is indeed owing to the correct prediction on the old class data.
>
> ------------------------------------
> (2). "To support the general overfitting claim made in section 4.1, the authors should repeat this analysis on any of the other five datasets where the performance difference between the two methods is much smaller.  SupportNet also suffers from overfitting (Table 3, Accuracy: test data: 83.9%, real training data: 98.7%) although to a lesser extent than iCaRL."
>
> Reply: Thank you for this nice suggestion. We repeated the same analysis on MNIST and added the result to Table 2. The result of MNIST shares the same pattern of the result on EC data. In terms of your comments on overfitting, we would like to explain a little bit more. As we discussed in Section 4.1, what we want to show in Table 2 is that the support data selected by our method are critical for the deep learning training. To show that, we reported the model's performance on 'All training data' (all the training data of all the time points that had once been fed to the model, including both all the old training data and the new training data) and 'Real training data' (the training data of the last time point, including the new training data and the support data) in Table 2. The results in Table 2 indeed support our claim, since the performance difference of SupportNet between 'Real training data' (98.7%) and 'All training data' (92.0%) is much small than that of iCaRL (99.1% and 62.6%). But you are right, SupportNet is not free of overfitting, which is the common problem of all deep learning based methods and should be handled with more efforts in the future.
>
> ------------------------------------
> (3). "The individual impact of the support points and the joint impact of support points with feature regularizer on accuracy is not assessed. To prove their usefulness, add two methods to Figure 3:
> (a)A method that uses support points without any regularizer.
> (b) A method that uses support points with just the feature regularizer. "
>
> Reply: Thank you for the very nice suggestion. We take the EC data as an example and added the comparison, which is shown in Fig. 3 (C). As shown in the figure, even with the support data along and without any regularizers, SupportNet can already outperform iCaRL on this dataset, which shows the effectiveness of combining SVM with deep learning to select the critical data points for training deep learning model in the continual learning scenario. At the same time, the two regularizers, feature regularizer and EWC regularizer, can improve the performance of SupportNet to different extents. Basically, the EWC regularizer has a larger impact on the SupportNet's performance than the feature regularizer. It is understandable since the EWC regularizer will influence each parameter individually according to their contribution while the feature regularizer tries to preserve the feature representations of the support data. Using feature regularizer along, the feature representations of the non-support data might be changed because of further training, which can have a negative impact on the model's overall performance. Since those two regularizers are responsible for different aspects in our framework, combining those two consolidation regularizers together with the support data, SupportNet can achieve the optimal performance.
>
> ---Thank you again and we will continue in the next post---

---

> > ### Author Response · Authors · 2018-11-16
> > **Response to AnonReviewer3--continued**
> >
> > Other points:
> > ------------------------------------
> > (1) "In section 2.3.2, EWC regularizer, Eq. 9: We think F(theta_new) should be F(theta_old) since we want to constrain parameters crucial for classification of old data and should be computing Fisher Information for the old parameters. "
> >
> > Reply: Thank you for pointing out the typo. We have corrected this in the revision.
> >
> > ------------------------------------
> > (2) "In section 2.1 Deep Learning and SVM: additional steps are needed to show how Eq. 3 is derived from Eq. 2."
> >
> > Reply: Thank you very much for raising this up. We added the derivation of Eq. 3 from Eq. 2 in Section A in the Appendix for the reader's reference.
> >
> > ------------------------------------
> > (3) "In section 2.1 Deep Learning and SVM: In the line before Eq. 4. “t represtent” instead of “t represents”.""
> >
> > Reply: Thank you for such detailed comments! We have corrected that in the revision.
> >
> > ------------------------------------
> > (4) "Figures are small and hard to read. Please increase the size and resolution of the figures."
> >
> > Reply: Thank you very much for your suggestion. We have increased the size and resolution of the figures.
> >
> > ------------------------------------
> > Thank you again for reviewing our paper. Please take a look at the updated version and please let us know if you have more comments.

---

### Author Response · Authors · 2018-11-16
**We have updated the paper and thanks for all the reviewers**

We thank all the reviewers for their time and comments!
We have uploaded a new version of our paper to address reviewers' comments. Here are the highlights of the changes:
1. We added the comparison of SupportNet with four more methods (shown in the below references) pointed out by the reviewers on the overlapped dataset, MNIST, whose result is shown in Fig. 3 (A), to support our main results.
2. We added the individual performance analysis on the old classes and the new classes with confusion matrices and accuracy matrices in Section B and Section C, to support our main results.
3. We further added the analysis of feature regularizer and feature representation learned by SupportNet using t-SNE, which is shown in Fig. 3 (C) and Section D, to support Section 2.3.
4. We added the performance of SupportNet with less support data size (1500, 1000, 500, 200) in Section E to further support Section 3.4.
5. We added the analysis on MNIST in Table 2 to further support Section 4.1.
6. We added the derivation of Equation 3 in Section A for reader's reference.
7. We would like to especially thank AnonReviewer3's detailed comments, pointing out some typos, which have been addressed in the updated version.

We also answered the reviewers' questions in the individual responses.
We would like to thank the reviewers once more for their valuable feedback. We hope they will find the changes satisfactory or we will wait for new feedback.

---

> ### Author Response · Authors · 2018-11-24
> **Another update**
>
> As suggested by reviewer 1, we have updated the manuscript with another experiment in Section F on a dataset with even more classes than CIFAR-100. We compared SupportNet with iCaRL on tiny ImageNet. The setting of tiny ImageNet dataset is similar to that of ImageNet. However, its data size is much smaller than ImageNet. Tiny ImageNet has 200 classes while each class only has 500 training images and 50 testing images, which means that it is even harder than ImageNet. The result is shown in Figure 10. As shown in the figure, SupportNet can outperform iCaRL significantly on this dataset. Furthermore, SupportNet's performance superiority is increasingly significant as the class incremental learning setting goes further, which, again, demonstrates the effectiveness of SupportNet in combating catastrophic forgetting.
>
> We would like to thank all the reviewers again sincerely, whose comments have definitely improved the quality of the SupportNet’s manuscript significantly.

---

### Meta-Review · Area_Chair1 · 2018-12-14

**Confidence:** 5
**Recommendation:** Reject

**Metareview:**

The authors propose using a SVM, trained as a last layer of a neural network, to identify exemplars (support vectors) to save and use to prevent forgetting as the model is trained on further tasks. The method is effective on several supervised benchmarks and is compared to several other methods, including VCL, iCARL, and GEM. The reviewers had various objections to the initial paper that centered around comparisons to other methods and reporting of detailed performance numbers, which the authors resolved convincingly in their revised paper. However, the AC and 2 of the reviewers were unconvinced of the contribution of the approach. Although no one has used this particular strategy, of using support vectors to prevent forgetting, the approach is a simplistic composition of the NN and the SVM which is heuristic, at least in how the authors present it. Most importantly, the approach is limited to supervised classification problems, yet catastrophic forgetting is not commonly considered to be a problem for the supervised classifier setting; rather it is a problem for inherently sequential learning environments such as RL (MNIST and CIFAR are just commonly used in the literature for ease of evaluation).